

# Integrated bioinformatics analysis of potential pathway biomarkers using abnormal proteins in clubfoot

Guiquan Cai[1], Xuan Yang[2], Ting Chen[2], Fangchun Jin[2], Jing Ding[2] and Zhenkai Wu[2]

[1] Department of Orthopaedics, Xinhua Hospital, School of Medicine, Shanghai Jiao Tong University, Shanghai, China
[2] Department of Pediatric Orthopaedics, Xinhua Hospital, School of Medicine, Shanghai Jiao Tong University, Shanghai, China

## ABSTRACT

**Background**. As one of the most common major congenital distal skeletal abnormalities, congenital talipes equinovarus (clubfoot) affects approximately one in one thousandth newborns. Although several etiologies of clubfoot have been proposed and several genes have been identified as susceptible genes, previous studies did not further explore signaling pathways and potential upstream and downstream regulatory networks. Therefore, the aim of the present investigation is to explore abnormal pathways and their interactions in clubfoot using integrated bioinformatics analyses.
**Methods**. KEGG, gene ontology (GO), Reactome (REAC), WikiPathways (WP) or human phenotype ontology (HP) enrichment analysis were performed using WebGestalt, g:Profiler and NetworkAnalyst.
**Results**. A large number of signaling pathways were enriched e.g. signal transduction, disease, metabolism, gene expression (transcription), immune system, developmental biology, cell cycle, and ECM. Protein-protein interactions (PPIs) and gene regulatory networks (GRNs) analysis results indicated that extensive and complex interactions occur in these proteins, enrichment pathways, and TF-miRNA coregulatory networks. Transcription factors such as SOX9, CTNNB1, GLI3, FHL2, TGFBI and HOXD13, regulated these candidate proteins.
**Conclusion**. The results of the present study supported previously proposed hypotheses, such as ECM, genetic, muscle, neurological, skeletal, and vascular abnormalities. More importantly, the enrichment results also indicated cellular or immune responses to external stimuli, and abnormal molecular transport or metabolism may be new potential etiological mechanisms of clubfoot.

Corresponding authors
Jing Ding,
dingjing@xinhuamed.com.cn
Zhenkai Wu,
wuzhenkai@xinhuamed.com.cn

## INTRODUCTION

As a common developmental malformation of newborns, clubfoot affects approximately 2% of newborns (*Wang et al., 2019*). If not actively and timely treated, this deformity will accompany the child for a lifetime, which will not only affect the appearance of the child, including walking difficulties or even a disability, but also cause serious adverse

effects on the mental health of the child. Therefore, timely and active treatments are urgently needed. The treatment of clubfoot includes surgical and nonsurgical treatments. Surgical releases, such as muscle balance and anterior tibial tendon transposition, may cause many problems, such as large trauma complications, which seriously affect the healthy development of children (*Smith et al., 2014*). The number of extensive surgical cases of clubfoot has declined by 60% from 1996 to 2006 in the United States (*Zionts et al., 2010*). In contrast, the Ponseti method has received increasing attention in the treatment of clubfoot and has become the most commonly used method. Although the success rate of Ponseti's first correction is satisfactory after a long period of development, there are still many problems and challenges, such as having a recurrence of deformity, having a poor compliance, and determining the best time to stop wearing braces (*Miller et al., 2016*; *Thacker et al., 2005*).

Clubfoot is associated with neuromuscular lesions, heredity abnormalities, skeletal dysplasia, soft tissue contracture, vascular abnormalities, ECM abnormalities, and intrauterine growth retardation (*Chesney, Barker & Maffulli, 2007*; *Eckhardt et al., 2019*; *Miedzybrodzka, 2003*; *Ošt'ádal et al., 2015*; *Poon, Li & Alman, 2009*; *Sodre et al., 1990*; *Wang et al., 2013a*; *Wang et al., 2013b*). In addition, smoking and viral infections in pregnant women are also closely related to congenital clubfoot (*Palma et al., 2013*; *Robertson Jr & Corbett, 1997*). Although the direct cause of congenital clubfoot remains to be unified, these findings provide an important opportunity and basis for the treatments of clubfoot. Several studies have revealed that botulinum toxin injection can relieve muscle or soft tissue contracture and may replace percutaneous tendoachilles tenotomy in the treatment of clubfoot (*Alvarez et al., 2009*; *Howren, Jamieson & Alvarez, 2015*). Based on the progress of pathogenesis, appropriate drug treatment may improve patient compliance and ultimately improve the efficacy of the Ponseti method. In addition, familial occurrence and inter- and intraphenotypic variability of clubfoot is well documented (*Basit & Khoshhal, 2018*). Several genes were identified as susceptible genes in clubfoot, such as the HOX family, CASP family, PITX-TXB4 pathway, troponin (TN) family, GLI3, T-box and MTHFR genes (*Hecht et al., 2007*; *Shrimpton et al., 2004*; *Shyy et al., 2009*; *Weymouth et al., 2016*; *Zhang et al., 2016*). However, these studies do not further explore abnormally active signaling pathways and potential upstream and downstream regulatory networks. Therefore, the aim of the present investigation is to explore the abnormal pathways and their interactions using integrated bioinformatics analysis in clubfoot. We expect the results of this study will provide an update on the etiopathogenetic mechanism of idiopathic clubfoot.

## MATERIALS AND METHODS

### Inclusion of abnormal proteins

Two widely used databases, PubMed and Science Direct, were used for literature retrieval. Keywords were ''(clubfoot or clubfeet or congenital talipes equinovarus) and (etiology or embryology or etiopathogenesis or genomics or genetics or pathology or pathophysiology)''. The search date was up to May 7, 2019. A total of 1,057 articles were found in PubMed, and 657 articles were found in Science Direct. After eliminating duplicate articles, a

total of 1,093 studies were retrieved. By reading the title and abstract, 1,015 articles were eliminated following the inclusion and exclusion criteria as described below. Inclusion criteria included clinical or preclinical studies written in English that were focused on the etiology of clubfeet. Investigations with a focus on secondary/syndromic clubfoot, such as distal arthrogryposis, myelomeningocele, and moebius syndrome, were excluded. We performed a full text assessment of the remaining 78 articles. A total of 47 articles were excluded because the study topic was not clubfoot, genetic information was not mentioned or full text of the study was not available. Finally, 8 articles were included in the present study. A total of 30 proteins that are shown in Table 1 were used in this investigation after assessing the protein. The process of the inclusion of abnormal proteins was shown in Fig. 1.

## WebGestalt analysis

WebGestalt analysis was performed as described previously (*Wang et al., 2017*). The parameters for the enrichment analysis were as follows. A specific organism was chosen *H. sapiens* (human). Enrichment categories were used KEGG, Panther and Reactome pathways. A reference list was used for all mapped entrezgene IDs from the selected platform genome. The reference list was mapped to 61,506 entrezgene IDs, and 2266 IDs were annotated to the selected functional categories and used as the reference for the enrichment analysis. The minimum number of IDs in each category was 5, and the maximum number of IDs was 2000. Among 30 unique entrezgene IDs, 19 IDs were annotated to the selected functional categories and used for the enrichment analysis. The Gene Ontology (GO) Slim summary was based upon 30 unique entrezgene IDs. Fisher's exact test-based overpresentation enrichment analysis (ORA) method was used for enrichment analysis. FDR was used for the Benjamani-Hochberg (BH) method.

## g:Profiler analysis

The version of g:Profiler was e94_eg41_p11_9f195a1 (database updated on 01/24/2019). The parameters for the enrichment analysis were as follows. A specific organism was chosen *H. sapiens* (human). GO analyses (GO molecular function (GO: MF), GO cellular component (GO: CC), and GO biological process (GO: BP)) were carried out sequentially. The biological pathways used were the KEGG, Reactome, and WikiPathways databases. The protein databases used were the Human Protein Atlas and CORUM databases. The statistical domain scope was used only for annotated genes. The significance threshold was the g:SCS threshold. The user threshold was 0.05.

## NetworkAnalyst analysis

The significantly changed genes from the previous analyses were mapped to the corresponding molecular interaction databases. The procedure typically produces one large subnetwork with several smaller ones. The website was upgraded and maintained until May 8, 2019 by the Xia Lab (https://www.networkanalyst.ca/NetworkAnalyst/faces/home.xhtml). The parameters for the enrichment analysis were as follows. A specific organism was chosen *H. sapiens* (human). The ID type was chosen Uniprot Protein ID. PPI analysis was performed using the MEx Interactome database. The parameters were referred

**Table 1  Protein information.**

| Entry ID | Entry name | Protein names | Gene names | References |
|---|---|---|---|---|
| Q9BXN1 | ASPN_HUMAN | Asporin | ASPN | *Ošt'ádal et al. (2015)* |
| P21810 | PGS1_HUMAN | Biglycan | BGN | *Ošt'ádal et al. (2015)* |
| Q8IUL8 | CILP2_HUMAN | Cartilage intermediate layer protein 2, CILP-2 | CILP2 | *Eckhardt et al. (2019)* |
| P35222 | CTNB1_HUMAN | Catenin beta-1 (Beta-catenin) | CTNNB1 | *Poon, Li & Alman (2009)* |
| P02461 | CO3A1_HUMAN | Collagen alpha-1(III) chain | COL3A1 | *Eckhardt et al. (2019)* |
| P20908 | CO5A1_HUMAN | Collagen alpha-1(V) chain | COL5A1 | *Eckhardt et al. (2019)* |
| P12109 | CO6A1_HUMAN | Collagen alpha-1(VI) chain | COL6A1 | *Eckhardt et al. (2019)* |
| Q99715 | COCA1_HUMAN | Collagen alpha-1(XII) chain | COL12A1 | *Eckhardt et al. (2019)* |
| Q05707 | COEA1_HUMAN | Collagen alpha-1(XIV) chain (Undulin) | COL14A1 | *Eckhardt et al. (2019)* |
| P08123 | CO1A2_HUMAN | Collagen alpha-2(I) chain | COL1A2 | *Eckhardt et al. (2019)* |
| P12110 | CO6A2_HUMAN | Collagen alpha-2(VI) chain | COL6A2 | *Eckhardt et al. (2019)* |
| P12111 | CO6A3_HUMAN | Collagen alpha-3(VI) chain | COL6A3 | *Eckhardt et al. (2019)* |
| P20849 | CO9A1_HUMAN | Collagen alpha-1(IX) chain | COL9A1 | *Wang et al. (2013a)* and *Wang et al. (2013b)* |
| O94907 | DKK1_HUMAN | Dickkopf-related protein 1, Dickkopf-1 | DKK1 | *Poon, Li & Alman (2009)* |
| Q06828 | FMOD_HUMAN | Fibromodulin | FMOD | *Eckhardt et al. (2019)* |
| P02751 | FINC_HUMAN | Fibronectin (Cold-insoluble globulin, CIG) | FN1 | *Ošt'ádal et al. (2015)* |
| Q14192 | FHL2_HUMAN | Four and a half LIM domains protein | FHL2 | *Wang et al. (2008)* |
| P35453 | HXD13_HUMAN | Homeobox protein Hox-D13 (Homeobox protein Hox-4I) | HOXD13 | *Wang et al. (2008)* |
| P51884 | LUM_HUMAN | Lumican | LUM | *Ošt'ádal et al. (2015)* |
| Q16853 | AOC3_HUMAN | Membrane primary amine oxidase | AOC3 | *Ošt'ádal et al. (2015)* |
| P20774 | MIME_HUMAN | Mimecan (Osteoglycin) | OGN | *Ošt'ádal et al. (2015)* |
| Q15063 | POSTN_HUMAN | Periostin (Osteoblast-specific factor 2) | POSTN | *Ošt'ádal et al. (2015)* |
| P51888 | PRELP_HUMAN | Prolargin | PRELP | *Ošt'ádal et al. (2015)* |
| Q7Z7G0 | TARSH_HUMAN | Target of Nesh-SH3 | ABI3BP | *Ošt'ádal et al. (2015)* |
| P24821 | TENA_HUMAN | Tenascin, TN (Cytotactin) | TNC | *Eckhardt et al. (2019)* and *Ošt'ádal et al. (2015)* |
| Q15582 | BGH3_HUMAN | TGF-β-induced protein ig-h3 | TGFBI | *Eckhardt et al. (2019)* |
| P48436 | SOX9_HUMAN | Transcription factor SOX-9 | SOX9 | *Wang et al. (2013a)* and *Wang et al. (2013b)* |
| P10071 | GLI3_HUMAN | Transcriptional activator GLI3 | GLI3 | *Cao et al. (2009)* |
| P01137 | TGFB1_HUMAN | TGF-β-1 proprotein | TGFB1 | *Wang et al. (2013a)* and *Wang et al. (2013b)* |
| P13611 | CSPG2_HUMAN | Versican core protein | VCAN | *Eckhardt et al. (2019)* and *Ošt'ádal et al. (2015)* |

to the literature-curated comprehensive data from the Innate DB (*Breuer et al., 2013*). Gene-miRNA interactome analysis was carried out with comprehensive experimentally validated miRNA-gene interaction data collected from TarBase and miRTarBase (*Chou et al., 2018*; *Vlachos et al., 2015*). TF-gene interaction analysis was performed using the ENCODE database. Transcription factor and gene target data derived from the ENCODE ChIP-seq data. The peak intensity signal <500 and the predicted regulatory potential

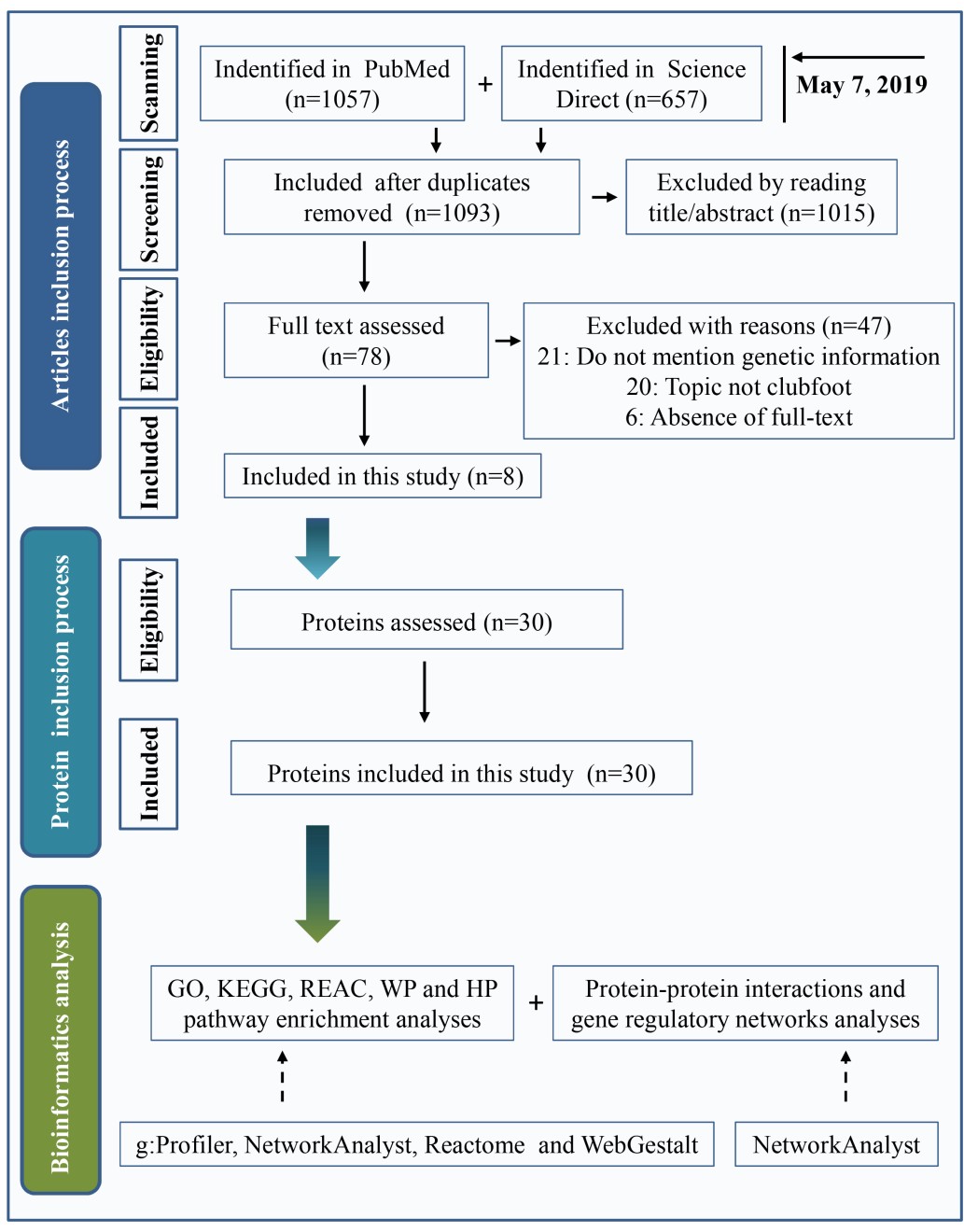

**Figure 1**  Inclusion criteria for abnormal protein candidates and the process of bioinformatics analysis.

score <1 used the BETA Minus algorithm (*Wang et al., 2013a*; *Wang et al., 2013b*). TF-miRNA coregulatory network analysis was performed using curated regulatory interaction information collected from the RegNetwork repository (*Liu et al., 2015*).

## REAC enrichment analysis

The enrichment analysis was performed against Reactome version 66 on 04/05/2019 using UniProt identifiers for the mapping. The web link is as follows: https://reactome.org/PathwayBrowser. Information in the REAC database is authored by expert biologists and entered and maintained by Reactome's team of curators and editorial staff. Reactome content frequently cross-references other resources, e.g., NCBI, Ensembl, UniProt, KEGG (Gene and Compound), ChEBI, PubMed and GO. REAC analysis was performed as described previously (*Fabregat et al., 2016*). The parameters for the enrichment analysis were as follows. A specific organism was chosen H. sapiens (human). IntAct interactors were included to increase the analysis background. An overrepresentation analysis method was used for enrichment analysis. This test produces a probability score, which was corrected for the false discovery rate using the BH method. Twenty-seven out of 30 identifiers in the sample were found in Reactome, and 430 pathways were found by at least one of the identifiers.

## Statistics

The enrichment analysis method in the WebGestalt analysis was used for ORA method. The significance threshold in the g:Profiler analysis was the g:SCS threshold (g:Profiler analysis soft version: e94_eg41_p11_9f195a1 (database updated on 01/24/2019)). The adjusted $p$ value method used was the BH method and the adjusted $p$ value was transformed to negative log10 ($-\log10$(AdjP)). All significantly changed pathways and interactions had a $p$ value <0.05.

# RESULTS

## Results of the enrichment analysis by g:Profiler and Reactome

To visually observe the enrichment information of these candidate proteins, g: Profiler and Reactome were performed for the bioinformatics analysis. A large number of terms were enriched by g:Profiler in the GO, KEGG, REAC, WP and HP databases (Fig. 2A). A large number of signaling pathways were enriched by Reactome (Fig. 2B). Additionally, there were extensive interactions among these signaling pathways (Fig. 2B).

To clarify the categories of these pathways, we summarized these pathways enriched by REAC. Signal transduction, disease, metabolism, gene expression (transcription), and immune system were the top 5 pathways and their percentages were as high as 65% (Fig. 3A). Among the 30 proteins, the top 10 proteins with the widest participation, such as P35222, P01137, P02751 and P08123, were mainly ECM proteins or proteins that interact with the ECM (Figs. 3B and 3C). We continued to summarize the top 10 pathways and found that these pathways were mainly focused on the ECM, metabolism and cell communication (Fig. 3D).

## Results of the signaling pathway enrichment analysis

Through the Reactome enrichment analysis, a large number of pathways were enriched (Figs. 2 and 3). To further analyze these potential pathways, signaling pathway enrichment analysis was carried out using g:Profiler in the REAC, WP, KEGG and HP. A total of 32, 11,
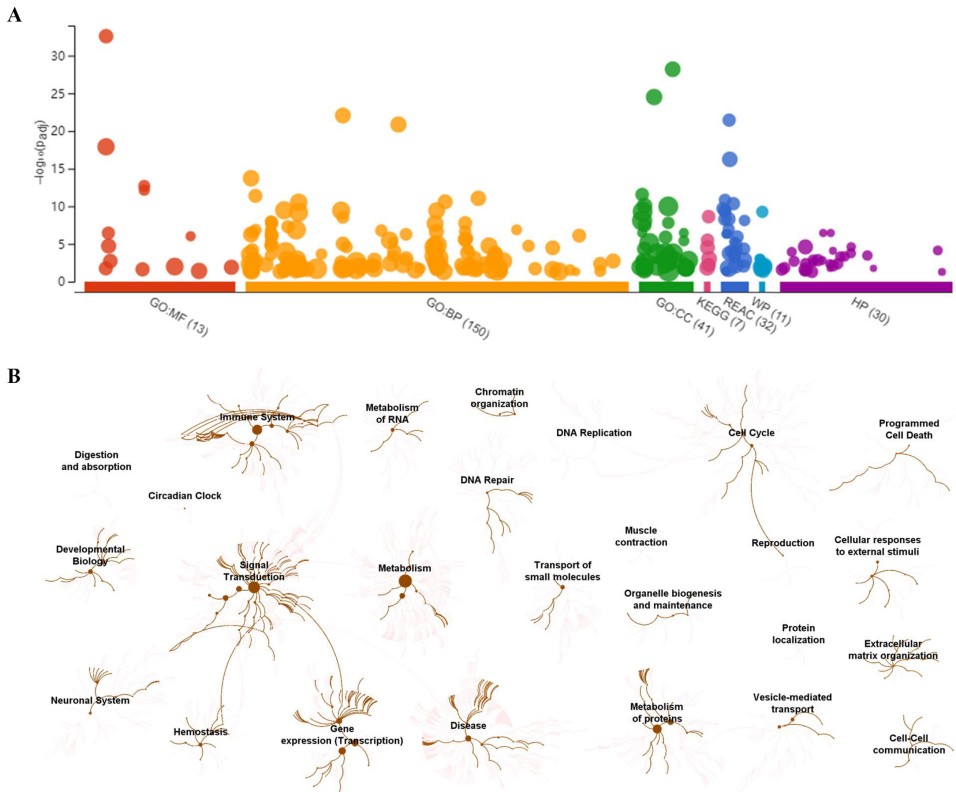

**Figure 2** **Overall results of bioinformatics analyses with candidate proteins using g:Profiler and Reactome.** (A) The significantly changed terms enriched by GO, KEGG, Reactome (REAC), Wikipathways (WP) and Human phenotype ontology (HP) databases. (B) The enrichment pathways and their interactions enriched by REAC.

7 and 30 pathways were enriched by g:Profiler, respectively (Figs. 4A–4D). Classification analysis revealed that these pathways were mainly concentrated in ECM, disease and metabolic pathways (Fig. 4E). The proportion of the top three was as high as 78%. In addition, human phenotype ontology enrichment analysis results revealed that these abnormal proteins were mainly expressed in the lower appendages, such as muscle, ankle, foot, joint, skin and connective tissue, and their proportion was more than 50% (Fig. 4F). Similar, similar results were found in KEGG and REAC enrichment analyses using NetworkAnalyst and WebGestalt (Fig. S1).

## Results of the GO enrichment analysis

GO enrichment analysis was further performed to explore the BP, MF and CC induced by these proteins. A total of 69, 9 and 24 GO terms were enriched by g:Profiler in BP, MF and CC, respectively (Figs. 5A–5C). Classification analysis revealed that GO: BP was mainly concentrated in embryo or organ development, and its combined percentage was over 80% (Fig. 5E). Among these pathways, skeletal development was dominant. ECM function, growth factor binding, cell adhesion, heparin binding and protein binding were the main pathways in MF, and their percentages were as high as 85% (Fig. 5F). Additionally, ECM

**A**

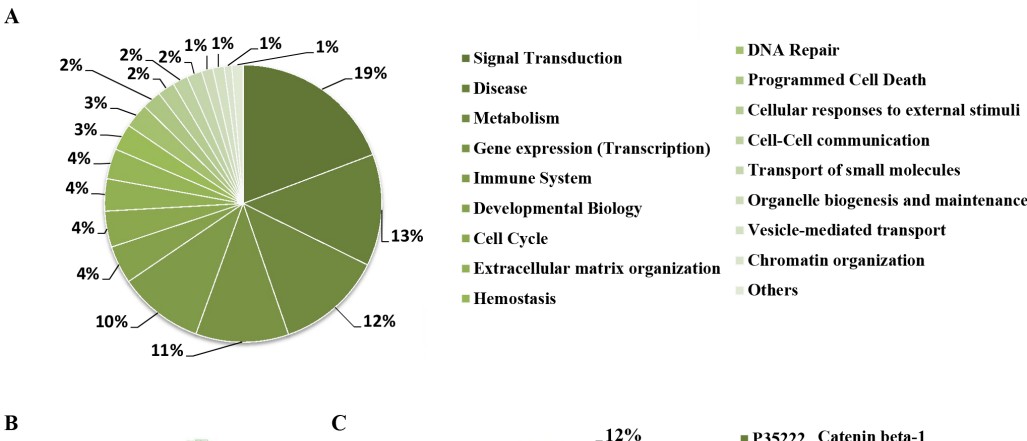

**B**     **C**

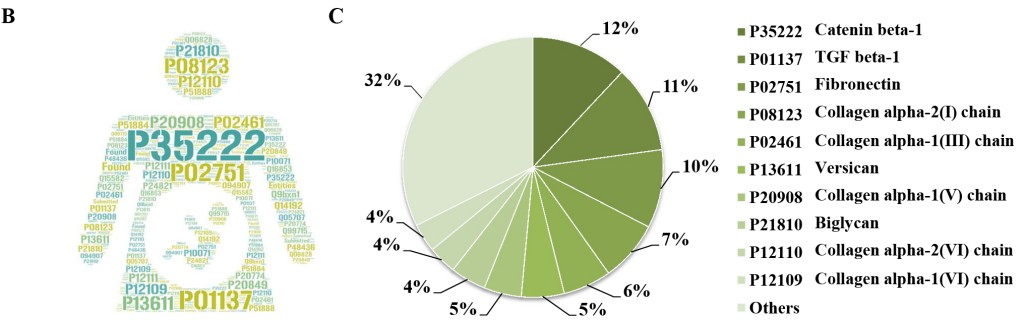

**D**

| FDR | Entities | Reactions | Interactors | Hits | Pathways | Type |
|---|---|---|---|---|---|---|
| | | | 5 | 1 | Activation of matrix metalloproteinases | ECM |
| 7 | 8 | 7 | | 3 | Assembly of collagen fibrils and other multimeric structures | ECM |
| | | | 8 | 1 | Cell-extracellular matrix interactions | Cell communication |
| 6 | 7 | 6 | | 3 | Collagen biosynthesis and modifying enzymes | ECM |
| 1 | 5 | | | 2 | Collagen chain trimerization | ECM |
| 2 | 6 | 8 | | 3 | Collagen degradation | ECM |
| 8 | 9 | 2 | | 3 | Collagen formation | ECM |
| 5 | 3 | 3 | 2 | 4 | Degradation of the extracellular matrix | ECM |
| 10 | | | | 1 | Diseases associated with glycosaminoglycan metabolism | Disease |
| 3 | 2 | | 10 | 3 | ECM proteoglycans | ECM |
| 4 | 1 | 1 | 1 | 4 | Extracellular matrix organization | ECM |
| | | 4 | | 1 | Glycosaminoglycan metabolism | Metabolism |
| 9 | 4 | 9 | 3 | 4 | Integrin cell surface interactions | ECM |
| | | | 9 | 1 | Localization of the PINCH-ILK-PARVIN complex to focal adhesions | Cell communication |
| | | 5 | | 1 | Metabolism of carbohydrates | Metabolism |
| | | | 6 | 1 | Non-integrin membrane-ECM interactions | ECM |
| | 10 | 10 | 4 | 3 | Signaling by receptor tyrosine kinases | Signal Transduction |
| | | | 7 | 1 | Syndecan interactions | ECM |
| | | | | | Degree | |
| 1 | 2 | 3 | 4 | | Hit numbers | |

**Figure 3** **Classification statistics of significantly enriched pathways by REAC shown in Fig. 2B.** Enrichment analysis methods are described in the Materials and Methods section. (A) Distribution of the enrichment pathways. (B) High frequency molecules in all of the significant signaling pathways were mapped by Wordart software (https://wordart.com). The larger the word frequency was, the larger the font size. (C) The top 10 proteins involved in enrichment pathways and their proportions. (D) The top 10 signaling pathways. Four columns on the left are the results of the top 10 FDR, entities, reactions and interactors. The numbers are their ranks. The hit numbers for each pathway are shown in column 5.

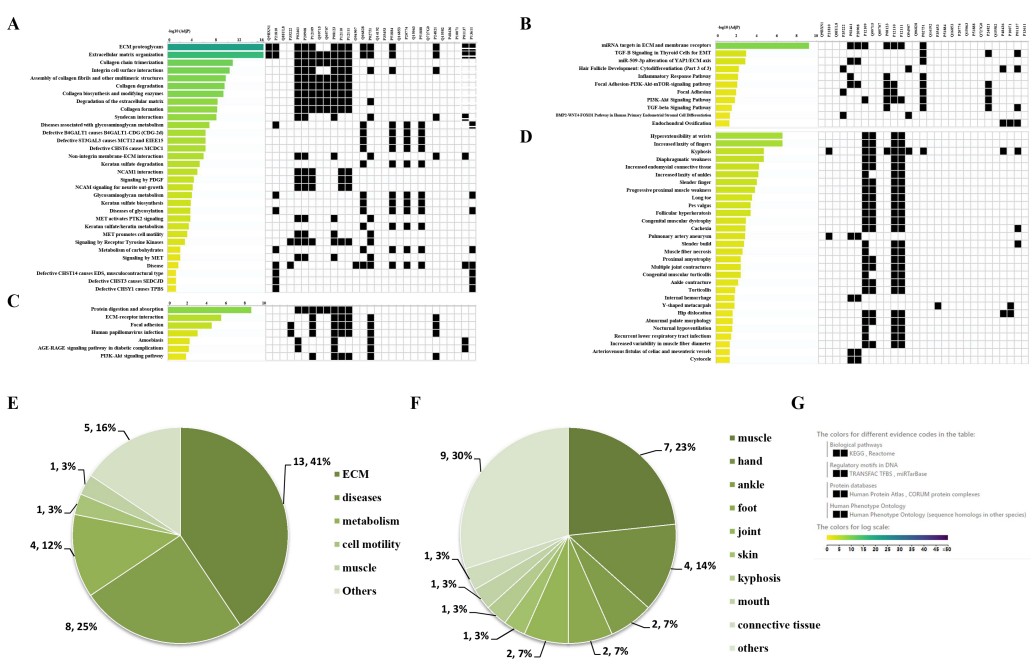

**Figure 4** **Signaling pathway enrichment analysis by g:Profiler in the REAC, WP, KEGG and HP databases.** Signaling pathway enrichment analysis by g:Profiler in the (A) REAC, (B) WP, (C) KEGG and (D) HP databases. Statistical results of signaling pathways from REAC (E) and HP (F). (G) Graphical illustration.

and membrane were the main CC, and their percentages were almost 60% (Fig. 5G). In addition, GO enrichment analysis results analyzed by NetworkAnalyst were consistent with those from g:Profiler (Fig. S2).

## Results of PPIs and gene regulatory networks analysis

The NetworkAnalyst analysis was performed to explore protein-protein interactions and gene regulatory networks for genes that code these candidate proteins. These genes interact extensively with predictive genes (Fig. 6A). A total of 2,630 PPIs were enriched by NetworkAnalyst. There are 1,263 interactions that were related to the top 10 genes and these interactions account for 48% of the total. The top genes were FN1, CTNNB1, FHL2, TGFB1 and COL1A2 (Fig. 6B). Among these genes, FN1 and CTNNB1 were dominant, and the percentage of PPIs induced by these two genes was almost 39%.

TF-gene interactions were also enriched by NetworkAnalyst. A total of 692 TF-gene interactions were enriched (Fig. 6C). The top genes were TGFB1, FN1, SOX9, AOC3 and HOXD13 (Fig. 6D). Additionally, the interactions among these 10 genes were as high as 43% of the total.

Gene-miRNA interactome analysis results revealed that 409 interactomes were identified between these genes and predictive miRNAs (Fig. 6E). The top genes were FHL2, PRELP, CTNNB1, SOX9 and COL12A1 (Fig. 6F). Top miRNAs were hsa-mir-124-3p, hsa-mir-26b-5p, hsa-mir-335-5p, hsa-mir-1-3p and hsa-mir-5698. The interactomes between the top 10 genes and the top 10 miRNAs accounted for 39% and 5% of the total, respectively.

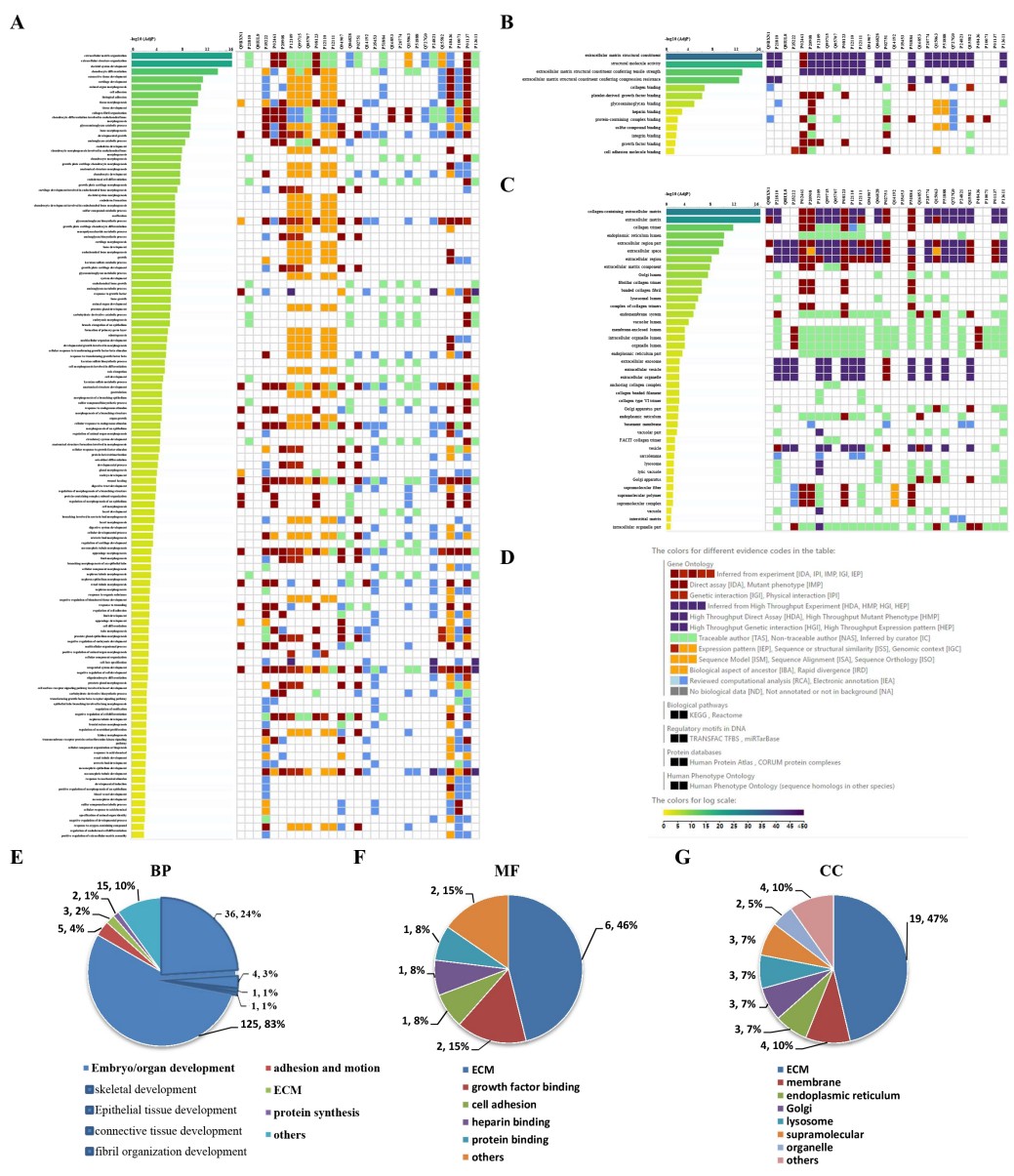

**Figure 5 GO enrichment analysis by g:Profiler: BP, MF and CC terms.** GO enrichment analysis by g:Profiler. (A) BP, (B) MF and (C) CC terms. (D) Graphical illustration. Statistical results from the GO BP (E), MF (F) and CC (G) enrichment analysis.

A large number of TF-miRNA coregulatory interactions were enriched (Fig. 6G). Transcription factors, such as SOX9, CTNNB1, GLI3, FHL2, TGFBI and HOXD13, cooperated with hsa-miR-29a, hsa-miR-101, hsa-miR-520d-5p, hsa-miR-29b and hsa-miR-568 to regulate these candidate genes (Fig. 6G). The top 10 genes were SOX9, CTNNB1, GLI3, FHL2, TGFB1, TGFBI, VCAN, TNC, COL12A1 and COL5A1 (Fig. 6H).

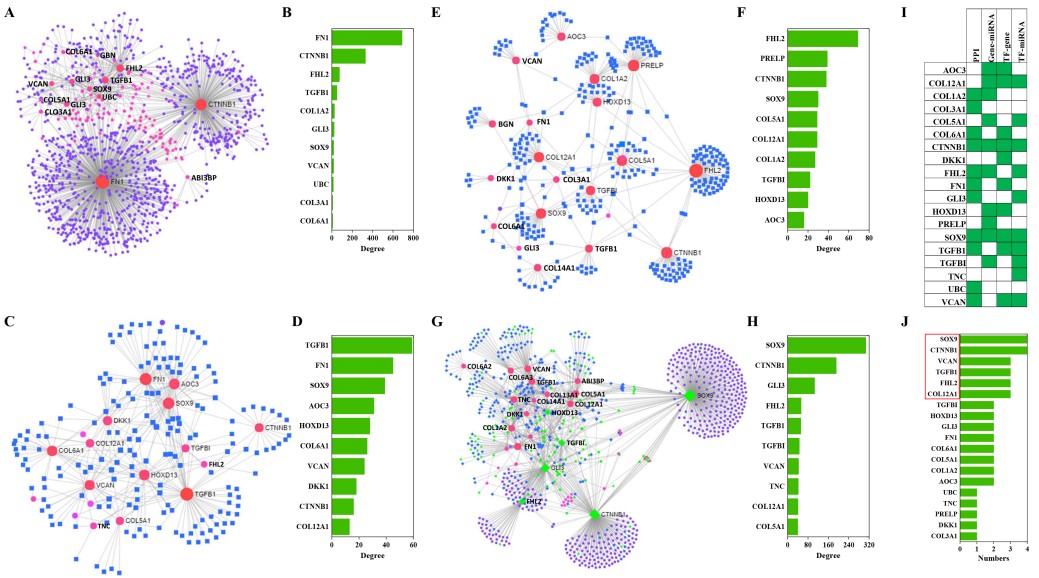

**Figure 6** **PPIs and GRNs analyses of these candidate protein-coding genes by NetworkAnalyst.** Generic PPI (A) and the top 10 genes (B) involved in that PPI. TF-gene interactions (C) and the top 10 genes (D) involved in these interactions. Genes are shown as pink circles, and transcription factors are shown as blue squares. Gene-miRNA interactome (E) and the top 10 genes (F) involved in these interactions. Genes are shown as pink circles and miRNAs are shown as blue squares. TF-miRNA coregulatory interactions (G) and the top 10 genes (H) involved in these interactions. Genes are circles. miRNAs are shown as blue diamonds. TFs are shown as green diamonds. The distribution (I) and frequency (J) of the top 10 genes in these four interaction networks mentioned above are shown.

Combining the four interactions mentioned above, 19 genes were involved in the top interactions (Fig. 6I). SOX9, CCNNB1, VCAN, TGFB1, FHL2 and COL12A1 widely participated in the top-ranking interactions (Fig. 6J).

## DISCUSSION

In the present investigation, a large number of signaling pathways were enriched in the GO, KEGG, REAC, WP and HP databases using g:Profiler, NetworkAnalyst and WebGestalt analyses of clubfoot. Among them, pathways in embryo or organ development, ECM, metabolism, immune system, cell cycle, cell responses to external stimuli, and apoptosis or programmed cell death were the top pathways. A wide range of interactions existed among these enriched signaling pathways. In addition, there were also extensive regulations between the upstream and downstream of genes encoding these proteins.

There were 452 enriched pathways identified by REAC enrichment analysis. Among them, signal transduction, disease, metabolism, gene expression (transcription), immune system, developmental biology, cell cycle, ECM and hemostasis were advantageous pathways. Additionally, signaling pathways in DNA repair, PCD, cell response to external stimuli, cell–cell communication, molecular transport and chromatin organization were enriched. In the process of cell biological activities, numerous changes occurred in cells, such as signal transduction, gene expression, cell cycle, DNA repair, molecular transport

and metabolism. Abnormal changes in these biological processes may alter cell fate, cell–cell communication, and cell response to external stimuli or even cause immune system changes.

Select soft tissues in clubfoot are contracted, resulting in stiffness. Extracellular matrix proteins, such as asporin, collagen types III, V, and VI, versican, tenascin-C, and TGF-beta induced protein, are highly expressed in contracted tissues clubfoot (*Eckhardt et al., 2019*; *Ošt'ádal et al., 2015*). Additionally, the expression levels of growth factors TGF-beta and platelet-derived growth factor are high and a blockade of growth factors led to decreased collagen expression, proliferation, and chemotaxis (*Li et al., 2001*). These proteins seem to be promising targets for future investigations and treatments of this disease. Indeed, the muscle contraction strategy, such as botulinum toxin injection, can relieve muscle or soft tissue contracture and thus improve and alleviate clubfoot symptoms (*Howren, Jamieson & Alvarez, 2015*; *Shrimpton et al., 2004*). ECM-associated pathways were enriched and included collagen chain trimerization, collagen formation and degradation, and ECM proteoglycans. TGF-beta receptor signaling was also enriched. These findings support the idea that these ECM proteins are promising targets for the treatment of clubfoot.

Abnormal biological development, such as muscle, neurological, skeletal and vascular abnormalities, has been previously identified and confirmed (*Basit & Khoshhal, 2018*; *Eckhardt et al., 2019*; *Herceg et al., 2006*; *Hester et al., 2009*; *Lovell & Morcuende, 2007*; *Ošt'ádal et al., 2015*). The HOX and TBX families governed limb identity, and fibroblast growth factor participated in the formation of limb muscles (*Ohuchi & Noji, 1999*; *Wang et al., 2008*). Additionally, gene-gene interactions between CASP SNPS and variants in HOXA, HOXD, and insulin-like growth factor binding protein affect muscle and limb development (*Ester et al., 2010*). Apoptosis and programmed cell death associated pathways and activation of HOX gene pathways, such as activation of HOX genes during differentiation, and activation of anterior HOX genes in hindbrain development during early embryogenesis, were enriched in the REAC database. A myogenesis pathway was also found.

ROBO family genes regulate axonal guidance and cell migration. ROBO1 and ROBO2 receptors regulate the proliferation and transition of primary to intermediate neuronal progenitors (*Borrell et al., 2012*), while the interaction of ROBO4 with SLIT3 is involved in the proliferation, motility and chemotaxis of endothelial cells, and accelerates the formation of blood vessels (*Zhang et al., 2009*). We found that the regulation of commissural axon pathfinding by SLIT, ROBO, netrin-1 signaling and pathways with signaling by ROBO receptors, semaphorin interactions, neurotransmitter release cycle were enriched. These results support that activation and inactivation of developmental signaling pathways initiate embryonic and organ development. Abnormalities in any link may cause deformities.

Epidemiological data confirmed that smoking by any parent or the presence of any household smoking increased the risk of clubfoot in Peru (*Palma et al., 2013*). Maternal diabetes also showed a significant association with clubfoot (*Parker et al., 2009*). In mice, maternal smoking or diabetes increased mitochondrial damage and oxidative stresses (*Stangenberg et al., 2015*). In addition, maternal diabetes also increased hypoxia-inducible factor 1 $\alpha$ expression in utero (*Moazzen et al., 2015*). *Robertson Jr & Corbett (1997)*

confirmed that clubfoot was caused by an intrauterine enterovirus through allowing anterior horn cell lesions. The adverse in utero environment caused by the external or internal environment may increase oxidative stress and thus induce cell damage in utero. We found that the signaling pathways involved in the regulation of gene expression by hypoxia-inducible factor, reactive oxygen species detoxification, cell response to hypoxia, cell response to heat stress, cell response to stress, oxidative stress-induced senescence, cellular senescence, and cellular responses to external stimuli were enriched in the REAC database. In addition, these external stimuli activate the immune system. We also found that interleukin family signaling, MAP kinase activation, toll-like receptor family cascade, adaptive immune system, and innate immune system were enriched. These data indicate that abnormal changes in the immune system and cell response to external stimuli may play key roles in clubfoot.

In the enrichment analysis, we also found that several molecular transport pathways were enriched by enrichment analysis, and induced plasma lipoprotein assembly, remodeling, and clearance, iron uptake and transport, small molecules transport, binding and uptake of ligands by scavenger receptors, lysosome vesicle biogenesis, vesicle-mediated transport, trans-Golgi network vesicle budding, and membrane trafficking. Molecular transport is essential for material transport, signal transduction and neurotransmitter transporters and modulates the cell response to an external force (*Hu & Papoian, 2013*). Disorders induced by abnormal molecular transport are serious and even fatal for cells. Based on these results, we proposed that molecular transport abnormalities may play a large role in clubfoot.

Metabolism of proteins, RNAs or other molecules plays a key role in the basic functions of cells (*Esteban-Martínez, Sierra-Filardi & Boya, 2017*). Pathways in these metabolic processes were found in the enrichment analyses, and induced regulation of IGF transport and uptake by IGFBPs, activation of genes by ATF4 in response to endoplasmic reticulum stress, synthesis, secretion, and inactivation of GLP-1, metabolism of nitric oxide, activation and regulation of eNOS, metabolism of carbohydrates, and regulation of lipid metabolism by PPAR $\alpha$. A previous study indicated that lipid droplets and glycogen increased and the number of mitochondria decreased in chondrocytes from the biopsied iliac crest cartilage of joint contracture patients (*Nogami et al., 1983*). These data suggested that glucose metabolism, lipid metabolism and mitochondrial oxidative stress were associated with multiple joint contractures. In the present work, glucose metabolism regulated by IGF or GLP, lipid metabolism regulated by PPAR, and oxidative stress regulated by NOS were found in the REAC database. These data suggest that metabolic abnormalities may play a significant role in clubfoot.

All candidate papers were obtained from PubMed and Science Direct. Almost all of these differentially expressed proteins were carefully and strictly selected from clinical trials. Clubfoot is abnormal in both bone and muscle. Most of the samples are from bone, and few are from muscle, therefore selection bias cannot be avoided. Through strict screening criteria, candidate proteins from different tissues were adopted for subsequent bioinformatics analysis, and the selection bias was possibly minimized. Although the adjusted $p$ value method was performed, there may be some false positives in using these candidate proteins for bioinformatics analysis because these proteins were not collected

from the same investigation. Although we proposed that cell or immune responses to external stimuli, abnormal molecular transport and metabolism are new potential etiopathogenetic mechanisms of clubfoot, direct experimental evidence is needed. We are carrying out preclinical and clinical studies to confirm these enrichment pathways and proposed hypotheses.

## CONCLUSIONS

In summary, a large number of signaling pathways were enriched using REAC, KEGG and WP enrichment analyses by g:Profiler, NetworkAnalyst and WebGestalt. Among them, signal transduction, disease, metabolism, gene expression (transcription), immune system, developmental biology, cell cycle, and ECM were the top functions. GO enrichment analysis also revealed pathways in embryo or organ development, and ECM proteins were dominant. PPIs and GRNs analysis results indicated that extensive and complex interactions occur in these proteins, enrichment pathways, and TF-miRNA coregulatory networks. Transcription factors such as SOX9, CTNNB1, GLI3, FHL2, TGFBI and HOXD13, cooperated with hsa-miR-29a, hsa-miR-101, hsa-miR-520d-5p, hsa-miR-29b and hsa-miR-568 and regulated these candidate proteins. In addition to supporting the proposed hypotheses, such as ECM abnormalities, fetal movement reduction, genetic abnormalities, muscle abnormalities, neurological abnormalities, skeletal abnormalities, uterine compression and vascular abnormalities, we propose that cellular or immune responses to external stimuli, and abnormal molecular transport or metabolism are new potential etiological mechanisms of clubfoot.

### Funding
The authors received no funding for this work.

### Competing Interests
The authors declare there are no competing interests.

### Author Contributions
- Guiquan Cai and Xuan Yang performed the experiments, analyzed the data, prepared figures and/or tables, authored or reviewed drafts of the paper, and approved the final draft.
- Ting Chen performed the experiments, analyzed the data, prepared figures and/or tables, and approved the final draft.
- Fangchun Jin analyzed the data, authored or reviewed drafts of the paper, and approved the final draft.
- Jing Ding and Zhenkai Wu conceived and designed the experiments, authored or reviewed drafts of the paper, and approved the final draft.

## Data Availability

The following information was supplied regarding data availability: Raw data is available in the Supplemental Files.

## Supplemental Information

Supplemental information for this article can be found online at http://dx.doi.org/10.7717/peerj.8422#supplemental-information.

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
