# Peer review of "Integrated bioinformatics analysis of potential pathway biomarkers using abnormal proteins in clubfoot"

_PeerJ, doi:10.7717/peerj.8422_

## Round 0.1 · original submission · Major Revisions

Our reviewers require further explanation of the rationale behind the culling of your search results to only eight papers. How did you ensure that no selection bias was introduced? I also agree with their assesment of the limited actionability of your insights, especially due to the lack of experimental validation, etc.

Reviewer 1 ·

Basic reporting

The expression of the article is basically clear, with some points needing polishing, such as line 144.The overall structure of the article is clear, many tables and figures are good for easily understanding.

Experimental design

1.The research idea of this article is simple and clear.However, the methods used in the 8 articles are different, and the location of the materials is also different. How much does this influence the experimental results?It needs to be explained in detail.
2.The principle and value of each step of the experimental method should be explained in more detail.

Validity of the findings

In this paper, only a few methods are used to obtain the possible important proteins. It would be more convincing if additional experiments were conducted to verify the results or the data from the database were still used for secondary verification.

Additional comments

The overall design of the article is novel and creative, clear results, sufficient analysis and discussion, detailed results are very valuable.However, the methods used in the 8 articles are different and the location of the materials is different, which should be explained.Secondly, it is better to increase the workload for the second verification of the results, so that the results are more convincing.

·

Basic reporting

The article written by the authors is very clear, unambiguous and technically, but there need more carefully in paper. It’s better to replace “lead to” with “including” or “such as” at lines 51, because there is no causality. “Ponseti” should be lowercase at lines 58 and 60.

Experimental design

1.PubMed and Science Direct described in the paper are commonly used by the bioinformatics community, but But why you don't try other databases? Such as TCGA and EBI. Please explain it.
2.It's better to supply a figure of PPI.

Validity of the findings

The problem addressed by the authors is important. Their key findings are beneficial to the medicine community to some extent.

Additional comments

No commens.

Reviewer 3 ·

Basic reporting

In this paper, the authors provided a bioinformatic analysis for identifying potential biomarkers for clubfoot. The paper need be revised before publication, especially in language. The expresssion could be improved. I suggest invite a native speaker to check the main text for avoiding grammer mistakes.

Experimental design

The experimental design of this paper is okay. While it is can be improved. For instance, the identified biomarkers are based on a single dataset. I suggest use another similar datasets to check the consistency in the experiments.

Validity of the findings

The findings of biomarker need be validated. For instance, the identified biomakers for classifying the samples. The application of biomarkers in this paper is for diagnosis. The reader might have the wonder of the classification performance of these identified biomarkers. I suggest add a section about the justification of these identified biomarkers.

Additional comments

The authors provided many GO enrichment fucntional analyses, such as the figures. However, it is still not clear the dysfunctions of the deformity. Moreover, the newly findings in genetics about clubfoot need be summarized in the descriptions about the search in PubMed. The figures need be presented in higher resolution. Many figures can not be recognized. Please also refer to my former comments for revising the paper.

Reviewer 4 ·

Basic reporting

Cai et al. present a bioinformatics analysis of genes implicated in developmental malformation (clubfoot). The authors performed a literature search, identified genes reported to be associated with the condition, and applied a range of bioinformatics approaches on those genes, including enrichment analysis, protein-protein interaction analysis, and others.

Shedding light on the pathogenesis of clubfoot is a worthwhile endeavor, given the high incidence of the condition. However, I have several major reservations on the manuscript:

- Goal of the project: It is not clear to me what the goal of the analysis is, and what actionable insights the large number of pathways that are predicted to be associated with the condition can provide. There is no overarching hypothesis that ties together the plethora of tools and approaches that were utilized.

- Approach: The authors performed a literature search to extract genes associated with the condition. However, the criteria that were used to retain or discard a PubMed or Science Direct entry are not specified. For example, on what grounds did the authors rejected 1015 articles? Study bias and selection bias are factors that need to be carefully considered here, especially when performing enrichment analysis.
It is also not clear what kind of associations were found between the genes they curated and the disease. Is it differential expression, mutation, or other?
Further, only a cursory description of the parameters used by each tool were given, and no rationale was provided for choosing a particular tool over another.

- Style: the manuscript has several typos throughout and many non-idiomatic expressions.

Experimental design

no comment

Validity of the findings

no comment

---

## Round 0.2 · accepted · Accept

Thank you for addressing our previous questions I look forward to seeing your manuscript in print.

Reviewer 3 ·

Basic reporting

In this version, the authors have addressed most of my former comments. For all the comments received, the authors have revised the manuscript accordingly. The identified pathway biomarkers are with the potential in development of clinical applications.

Experimental design

The authors searched in literature to collect the genes related to clubfoot. Then the paper implements enriched analysis to identify pathways. Then the biomarkers have been identified by classification.

Validity of the findings

The authors need strengthen the justification of these novel biomarkers. This is very important in biomarker discovery. The authors need play attention to this point.

Additional comments

In this version, the authors have addressed most of my former comments. I appreciate the responses. I think this version is much better than the former one. I support its publication. The authors should review the pathway biomarkers because this is an important concept. Moreover, the validation the identified biomarkers is very important in clinics.

Reviewer 4 ·

Basic reporting

no comment.

Experimental design

no comment.

Validity of the findings

no comment.